# *Salmonella* Control in Swine: A Thoughtful Discussion of the Pre- and Post-Harvest Control Approaches in Industrialized Countries

**DOI:** 10.3390/ani14071035

**Published:** 2024-03-28

**Authors:** Ana Carvajal, Melvin Kramer, Héctor Argüello

**Affiliations:** 1Departamento de Sanidad Animal, Campus Vegazana, 2007 Leon, Spain; ana.carvajal@unileon.es; 2EHA Consulting Group, Fort Lauderdale, FL 33306, USA; mkramer@ehagroup.com

**Keywords:** pig, zoonosis, public health, foodborne pathogen, mitigation

## Abstract

**Simple Summary:**

Zoonoses, particularly foodborne zoonoses, are a major problem worldwide, jeopardizing human and animal health and negatively impacting other aspects such as the economy, food systems, and human and animal wellness. This review focuses on the control of the foodborne pathogen *Salmonella* in pigs and pork, revising past and ongoing control programs, the control options in the food chain, and the impact of policies implemented worldwide. The objective is to take a closer look at the efforts made in the last 2–3 decades in *Salmonella* control in pigs, evidencing their strengths, pitfalls, and limitations and determining future directions for the efficient surveillance and control of this pathogen in the future.

**Abstract:**

Pork is among the major sources of human salmonellosis in developed countries. Since the 1990s, different surveys and cross-sectional studies, both national and international (i.e., the baseline studies performed in the European Union), have revealed and confirmed the widespread non-typhoidal *Salmonella* serotypes in pigs. A number of countries have implemented control programs with different approaches and degrees of success. The efforts could be implemented either at farms, in post-harvest stages, or both. The current review revises the current state of the art in *Salmonella* in swine, the control programs ongoing or conducted in the past, and their strengths and failures, with particular attention to the weight of pre- and post-harvest control and the implications that both have for the success of interventions or mitigation after outbreaks. This review provides a novel perspective on *Salmonella* control in swine, a matter that still includes uncertainties and room for improvement as a question of public health and One Health.

## 1. Introduction Scope and Aim of This Review

As a pathogen, *Salmonella* is a genus in which species and serovars have evolved to colonize different hosts and environments [1]. Among the serovars included in *Salmonella enterica*, subspecies *enterica*, a few have adapted to specific hosts, causing severe disease. For instance, host-specific serovars include *Salmonella* Typhi and *S.* Paratyphi in humans and the serovar *S.* Cholerasuis in pigs [2,3,4]. In contrast to these host-adapted serovars, the vast majority of *Salmonella* serovars have chosen a different evolutionary strategy, allowing them to colonize a wide range of hosts, including warm- and cold-blooded species. They are generically known as non-typhoidal serovars, a definition that refers to their tendency to restrict colonization to the intestine without causing systemic disease in the host, unlike the aforementioned serovars [3].

Over the last half-century, non-typhoidal serovars have successfully spread into food production animals, particularly in pig and poultry production. Industrialized production systems with high stock densities, coupled with subclinical outcomes of infection characterized by large pathogen shedding in feces with mild or no pathogenic signs, have facilitated the spread of *Salmonella* in poultry, turkey, pig, and calf productions [5,6,7,8,9].

Consequently, *Salmonella* is a major foodborne pathogen, causing approximately 150 million illnesses and 60,000 deaths globally each year [10]. Major sources of *Salmonella* infection include chicken, turkey meat, eggs, pork, and derived products [11,12]. Human salmonellosis, attributed to pork consumption, ranks among the most highly reported foodborne illnesses [12]. Since the 1990s, many countries have implemented control and surveillance programs aimed at reducing the risk of *Salmonella* transmission by pork into the food chain [13,14,15,16]. These programs target *Salmonella* control at different production stages, including feed mills, farms, and post-harvest control. Each strategy or approach has its strengths and shortcomings, leading to different perspectives among the main actors involved in pathogen control. This review aims to bring the readers closer to the control of *Salmonella* in pig production tackling aspects of the origin of control programs, their progression or even discontinuation, main aspects for pre and post-harvest control and legal aspects of *Salmonella* outbreaks in humans in different countries. This review provides a novel perspective on reviews focusing on *Salmonella* control in swine, an issue that still includes uncertainties and room for improvement in public health and One Health.

## 2. Identifying Relevant Literature

This study did not apply a systematic review or meta-analysis. Instead, the relevant literature on the different topics covered in the review was searched using different scientific databases (www.pubmed.com; accessed the last ie on 20 January 2024; www.sciencedirect.com; accessed the last ie on 20 January 2024) and the website search tool (www.google.es; accessed the last ie on 20 January 2024). Relevant scientific papers published in peer-reviewed English journals were identified using the following keyword combinations: (pig OR Salmonella OR farm OR slaughterhouse OR control OR program) AND (livestock OR swine OR farrow OR weaner OR finisher OR sow OR carcass) AND (acid OR vaccine OR antimic* OR HACCP, OR risk), among others. The search did not have any restriction on dates, so any document in the databases to date was searched to capture up-to-date data. To ensure a wide range of articles from different sources, additional searches were conducted using the reference lists of key articles.

## 3. *Salmonella* Control in Swine Production: Origins

*Salmonella* control in swine production was initiated in the 1990s and early 2000s by the establishment of national initiatives to reduce the burden of this pathogen in the food chain, encouraged in part by the results of programs in poultry production.

The pioneering program was the Danish *Salmonella* control program, established in 1993, with the development and implementation of a surveillance program for Danish pork and for slaughter pig herds in 1995 [13]. The program was a response to human cases of salmonellosis linked to pork, with a peak incidence in 1993 [17,18]. Other programs in Europe followed the Danish initiative. Chronologically, Ireland established a surveillance program for pig herds [14] in 1997, which remains in force today [19]. The German *Salmonella* Monitoring Program was established in 2002 through the German Quality Assurance System for the food chain, the so-called “QS-System” [20,21]. In the same year, the pig industry in the UK launched the Zoonosis Action Plan (ZAP) to categorize pig herds based on their *Salmonella* prevalence. The program was revised in 2008 under a new name, the “Zoonoses Control Program” [16], and was finally suspended in 2012 [22]. Finally, the Netherlands and Belgium established their respective programs in 2005. In the Netherlands, compulsory *Salmonella* monitoring in fattening pigs was initiated by the Product Boards for Livestock, Meat, and Eggs, whereas the Belgian program [23], named the National *Salmonella* Action Plan (SAP), originally targeted herds with more than 30 pigs and has since changed several times, currently operating on a voluntary basis [24]. For further information about control programs in European countries, specific reviews are accessible elsewhere [25].

Outside of European countries, few have undertaken actions to mitigate the pathogen in primary production. Despite the extensive literature on *Salmonella* epidemiology in Asian countries [26,27,28] and North and South American regions [29,30,31,32], only specific actions have been implemented in the United States [33]. In this country, the Food Safety and Inspection Service (FSIS) published a final rule on pathogen reduction (PR) and hazard analysis and critical control point (HACCP) systems in 1996. The final rule required meat and poultry establishments under federal inspection to take responsibility for preventing and reducing physical, chemical, and biological hazards throughout the food production process by implementing a system of science-based preventive controls, known as HACCP. Establishments must have an effective HACCP food safety system to comply with regulatory requirements, focusing on controlling hazards to prevent product adulteration. This approach relies on post-harvest decontamination for *Salmonella* control, with no surveillance in primary production. Undoubtedly, the approach taken to perform *Salmonella* control impacts the results and approaches to dealing with the pathogen, factors which will be introduced and discussed in subsequent sections.

## 4. Rationale of *Salmonella* Surveillance in Control Programs

*Salmonella* control can be approached from various perspectives or strategies, all valid but each with its weaknesses. Not previously mentioned, the most successful strategy, followed in Sweden, Norway, and Finland, is based on pre-harvest surveillance programs combined with an eradication strategy [34,35,36]. This strategy maintains *Salmonella* prevalence at 0% by identifying infected animals that are then condemned. However, this ideal strategy is only feasible when sporadic infections in production animals occur. In countries with a significant prevalence or larger production, characterized by higher movement and pig imports, this approach would fail, and the cost of eradication would be impractical. Therefore, in countries with a non-negligible herd and within-herd prevalence of *Salmonella*, more realistic options are needed to mitigate the risk of human infections.

All other European control programs rely on farm categorization based on serological *Salmonella* surveillance (Table 1). This on-farm categorization is based on the presence and quantification of antibodies against *Salmonella*-LPS, which demonstrate the on-farm contact between the pathogen and the animal [37]. Serology through Enzyme-Linked Immunosorbent Assays (ELISA) has been the technique of choice for several reasons. Firstly, it is less expensive than microbiological methods, can be easily automated, and *Salmonella* antibodies persist for long periods, overcoming the intermittent fecal shedding that may lead to false-negative results. Additionally, samples can be easily collected at the slaughterhouse (either sampling blood or meat juice samples). Thus, this technique has become the gold standard for *Salmonella* monitoring, with only the Danish control program conducting complementary bacteriological analyses [18].

Despite herd surveillance and on-farm categorization based on *Salmonella* indirect burden estimation, not all programs act based on prevalence reduction or penalty systems considering their *Salmonella* serological results (Table 1). Indeed, only the Danish and German programs have established penalty systems that devalue carcass refunds from animals from infected farms [18,25]. Furthermore, some programs (such as in Denmark) also monitor carcass prevalence. Danish prevalence targets are based on carcass prevalence rather than herd prevalence [18]. *Salmonella* carcass prevalence provides complementary information to serological surveillance on the farm, reflecting the potential introduction of *Salmonella* into the food chain (cutting plants and retailers). Moreover, this relevant information can be used to assess the effectiveness of interventions taken to reduce prevalence, not only on the farm but also in post-harvest stages [38]. Carcass monitoring is also performed in EU countries, regardless if they have a control program or not, by the Competent Authorities and Food Business Operators, either by official requirements or internal audits. Although this carcass monitoring is not part of the control programs, the results may help to inform about the success of the interventions put in place. More information about carcass contamination and results from scientific studies can be found in specific reviews on the topic [38,39]. The national control programs currently underway do not establish penalties for either the supplier (farm) or processor (abattoir) associated with carcass prevalence, although purchasers or importers may establish regulations that include the absence of *Salmonella* in the products, thus establishing an indirect firewall against *Salmonella*-contaminated pork.

The aim of these programs is not just limited to detecting *Salmonella* carriers in the food chain but also to reducing the *Salmonella* burden over time. Have the current control programs and their policies achieved any reduction? This question will be discussed in the next section.

## 5. Outcomes of over Two Decades of *Salmonella* Surveillance

The ongoing programs mentioned above imply a significant economic investment and efforts by public bodies involved in official control, as well as industries, practitioners, and farmers participating in the surveillance. Indirectly, concern about *Salmonella* also generates costs in research about *Salmonella* etiology, epidemiology, and control. With this information in hand, the question to be asked is probably “is all this effort worthwhile?”.

A case study published by Alban and colleagues in 2012 [18] described the lessons learned in the Danish *Salmonella* surveillance and control program for pigs and pork. In 2001, its prevalence in pork produced by members of the Danish Agriculture & Food Council was 1.7%, according to that study. A decade later, in 2011 and onwards, carcass prevalence ranged between 1 and 1.4% [40]. According to the authors of the last reference, in 2015, 24 years after the beginning of the program, the Danish *Salmonella* program reduced the number of human cases associated with Danish pork by more than 95% [40]. Undoubtedly, these efforts show figures that highlight the benefits of the interventions. But what are the costs? According to Danish researchers, the initial expenses of the program were approximately EUR 13 million per year. Adjusting the program, costs were significantly reduced to approximately EUR 3 million per year. To be a pioneer implies risks and failures. Not everything in the Danish program is a successful story, and some interventions or decisions did not provide the desired results. Between 1996 and 2010, an eradication policy was pursued for a highly concerning strain from a human health perspective, the multi-resistant *Salmonella* Typhimurium DT104 [41]. The plan did not achieve the established targets, and the total expenses spent were enormous, reaching more than EUR 14 million [18]. The nature of the design made *Salmonella* TyphimuriumDT104 vanish from pig farms in the decade of the 2010s, replaced by other emerging strains [42,43].

The demonstrated success of the Danish control program, both in carcass prevalence and human-related cases, contrasts with the partial figures reported by other European control programs. The ZAP program in the UK, as mentioned above, was discontinued in 2015. The German control program aimed at reducing herds with high seroprevalence of *Salmonella*. According to data provided by Blaha and colleagues [44], although herds falling into the highest level category of *Salmonella* prevalence were able to reduce prevalence figures, thus lowering their intra-herd prevalence of *Salmonella*-antibody positive pigs and herds with mild or low prevalence used to increase in prevalence. As a result, the overall national frequency of *Salmonella*-antibody-positive pigs supplied to the slaughter plants did not show any sort of improvement [45]. Other programs such as the Irish or Belgian programs also did not observe any improvement in *Salmonella* figures [21]. Consequently, the Irish NSCP is under review by the Pig HealthCheck program of Animal Health Ireland [25].

Why have these programs failed? Probably, the question requires a complex analysis, but several reasons can be identified as major motives. Firstly, as mentioned above, different programs such as the Belgian, Irish, or Dutch do not include penalties for high-prevalence herds. Without that penalty, the motivation to reduce on-farm prevalence may not be as crucial. A recent study performed in Denmark observed differences among farmers’ perceptions of costs and actions toward *Salmonella* control, which might hamper the effectiveness of the penalty scheme as a regulatory instrument to influence farmers’ behavior [46].

Other potential challenges that may hamper the interventions taken could be related to the complex control of the pathogen [47,48,49,50], as well as human factors [51]. For instance, Blaha identified a “lean-back” attitude in farmers that succeeded in achieving a reduction in *Salmonella* prevalence, linking low prevalence serology with *Salmonella*-free herds, while the pathogen was still circulating there. That relaxation in the interventions taken to reduce *Salmonella* prevalence ended in an increase in prevalence finally [44]. A similar idea is reflected by Marier and colleagues in a study performed in the UK [52]. This study confirmed that farmers accepted their responsibility for controlling *Salmonella* in pork, even though their confidence in their ability to control *Salmonella* decreased over time, and believed that responsibility should be shared with the rest of the production chain, a fact that matches the perception of German studies [45].

The control of the pathogen seems to be complex according to the information gained so far in this article. However, complexity is not synonymous with impossibility. The next sections summarize the experience gained in *Salmonella* control both on farms and in post-farm stages.

## 6. *Salmonella* Control in Pig Production

To assist surveillance programs in pig production, actions must be taken to mitigate *Salmonella* risk throughout the food chain. *Salmonella* mitigation strategies can be approached from different angles, meaning different production stages. Each intervention will have benefits but also weaknesses and risks of reinfection or recontamination. In this subsection, we aim to discuss the various alternatives available for *Salmonella* control (Figure 1). A thorough review of potential interventions at each production stage is not the objective of this review. Detailed and extensive reviews on this topic can be found elsewhere [39,44,47,48,51,52,53,54,55,56,57]. Nonetheless, several notes about the two main stages of pathogen control will be described in the subsequent paragraphs. This information will allow us to discuss the impact of on-farm and post-harvest control strategies for the control programs and the perception of the actors involved in *Salmonella* control.

### 6.1. The Control of Salmonella on the Farm

The study of control options to mitigate *Salmonella* prevalence and infection transmission on swine farms has been and is the subject of scientific research. Numerous published scientific peer-reviewed articles have evaluated in vivo control strategies or risk factors for *Salmonella* prevalence on swine farms, all aiming to extend the current knowledge in *Salmonella* control and identify tools and strategies to mitigate the risk of infection or the burden of this pathogen in live pigs. Despite the extensive research, variability in results is common, thus not providing absolute certainty about the formula to mitigate *Salmonella* on the farm.

In 2015, the FAO published a systematic review of *Salmonella* control in beef and pork, covering different steps in the food chain [58]. The FAO book report performed a thorough review of the options to control the pathogen on farms. Among all the interventions mentioned, experts highlighted several as the most efficient or useful. Biosecurity was highlighted as a good farm practice limiting the likelihood of pathogen introduction on the farm. Several studies have demonstrated that improved on-farm biosecurity limits the risk of on-farm *Salmonella* prevalence [59,60,61]. It seems logical to think that it is more plausible to introduce *Salmonella* by “four legs” than other potential sources such as feed, wild animals, or through water or semen. Nonetheless, all can be prevented with appropriate measures of external biosecurity on the farm, particularly restrictive policies for vehicles and visitors who come into contact with other herds. Measures of internal biosecurity, meaning limiting the transmission or circulation of the pathogen within the farm, seem to be less efficient than external biosecurity measures, probably due to lower or poorer internal biosecurity [32,62]. All-in/all-out husbandry, which includes emptying and cleaning barns and disinfection of facilities before introducing new animals, is the most useful tool to break infection cycles between animal batches [63]. Despite this activity being common among herds, risk factor studies have not demonstrated any clear benefit for *Salmonella* mitigation when this intervention is applied [59,64,65]. Different aspects like inefficient cleaning or commingling of animals from different batches may be behind the lack of benefits [66]. Thus, while introducing the pathogen on a farm can be prevented, once inside, breaking transmission is highly difficult.

The FAO document also mentions other interventions associated with feed or vaccination [58]. Feed serves as a vehicle for different strategies to mitigate *Salmonella* on the farm. Feed form [49,67,68], feed coarseness [69], feed additives [70,71,72], or probiotics administered through feed [73] have demonstrated their potential beneficial effect either by impacting the pathogen directly [74] or by improving intestinal health, thus limiting the opportunity for *Salmonella* to colonize the gut. Apart from the mentioned strategies, only vaccination is a strategy that can be performed on a large scale [58]. Again, different vaccines have been tested using different vaccination approaches [75,76,77,78]. *Salmonella* vaccines are not among the commonly used vaccines on farms, probably for different reasons; firstly, because they are not available in all countries, and secondly, because their efficacy is limited to the serotype infecting the animal [77]. While countries with serological surveillance may be reluctant to the use of vaccination due to interference with serological tests, they are extensively used in countries with clinical problems of salmonellosis like the USA (Fernando Leite, personal communication). Current policies to reduce antimicrobial use, together with the therapeutic use of ZnO in the EU [79], may be a reason to increase the use of vaccines in *Salmonella* control.

Apart from these interventions, other options are currently at a lower technology readiness level. For instance, phages, new nutraceuticals, or antimicrobial peptides are strategies with potential but with low field applicability right now [80,81,82]. There is not a magic formula for *Salmonella* control on the farm. Indeed, none of the options mentioned in this section exhibits enough robustness to successfully control the pathogen on its own. Controlling *Salmonella* on farms involves a combination of strategies, and their efficacy can vary based on farm-specific factors, management practices, and the *Salmonella* serotype involved. In a recent study evaluating factors associated with high and low seroprevalence on Irish farms, based on surveillance data from the National Control program, we demonstrated that low prevalence herds were related to farms in which aspects associated with biosecurity, feed coarseness, and herd health influenced the on-farm *Salmonella* prevalence [49].

In summary, a comprehensive and multifaceted approach, incorporating various control measures, is most effective in managing *Salmonella* on farms. The combination of biosecurity, sanitation, water and feed management, pest control, vaccination, monitoring, and employee training is key to reducing the prevalence and impact of *Salmonella* infections on pig farms.

### 6.2. Salmonella on Post Farm Stages, Epidemiology and Control

Post-harvest is the term that defines the production stages after the farm and which include transport of the animals to the slaughterhouse, their resting, slaughter and carcass fabrication and the latter stages of pork meat processing at the cutting plants and retailers. Again, the number of studies that have focused their aims on disclosing information about the dynamics of *Salmonella* in these stages and the potential mitigation strategies to be put in place is almost overwhelming. If readers want to extend their knowledge about *Salmonella* and post-harvest in pig production, we recommend a few review studies which go into depth into this topic [39,56,83,84,85].

There are a few studies that exemplify what happens in the *Salmonella* epidemiology after the farm. In 2001, Hurd and colleagues [86] performed a study in which 50 pigs were slaughtered on the farm and another 567 market-weight pigs were transported (mean distance, 169 km) to the slaughterhouse with 2 h to 3 h of holding in ante-mortem pens before slaughtering, as usual. Lymph nodes from both groups were collected and *Salmonella* prevalence was determined by microbiological detection of the pathogen. Interestingly, the prevalence at the slaughterhouse was five times higher than on the farm. In 2013, the European Food Safety Agency (EFSA) published the results of the analysis performed on carcasses from finisher pigs, as part of the cross-sectional study performed in member states to determine the basal prevalence of *Salmonella* in slaughter pigs [87]. From the results and conclusions obtained in the study, it is noteworthy to mention here that carcass prevalence was positively associated with on-farm prevalence, i.e., the higher the prevalence on the farm, the larger the number of contaminated carcasses detected. However, when farms with similar prevalence supplied different abattoirs, differences in carcass prevalence were observed among slaughterhouses. The third study to mention is a study in which the authors aimed at organizing the slaughtering by means of on-farm seroprevalence [88]. First, pigs from low prevalence farms were slaughtered followed by farms with high prevalence. Unexpectedly, when carcass prevalence was analyzed, a higher prevalence was obtained in carcasses from *Salmonella*-free and low *Salmonella* seroprevalence herds compared to high seroprevalence herds. Further typing of *Salmonella* isolates obtained from concomitant samples collected in the slaughterhouse environment revealed the link between carcasses from *Salmonella*-free herds and lairage environment, thus revealing new infections or contamination occurring in the slaughterhouse facilities.

We have chosen these three examples as they exemplify with accuracy the role of post-harvest in *Salmonella* epidemiology and control after the farm. The study of Hurd and colleagues, supported by results from subsequent studies [89,90,91,92], clearly highlights the potential hotspots in *Salmonella* new infections or re-infections after the farm. Indeed, the studies referenced and others [39] point out both transport and lairage as hotspots where new infections can occur and also where re-activation of infections can happen. The arrival of pigs from infected herds which spread *Salmonella* in these environments, sometimes in large concentrations, together with the rapid infection onset and fecal shedding in infected animals [93], explains the results observed by Hurd and colleagues in their study performed two decades ago. Is this information asserting that post-harvest is more important in *Salmonella* control than the on-farm prevalence?

The baseline *Salmonella* prevalence study performed in the EU and already mentioned [87,94] is the largest study performed so far, all over the world, involving a total of 19,159 slaughter pigs with validated results from 26 countries and monitored 5736 carcass swab samples in 146 slaughterhouses (from 13 Member States). The weighted prevalence of *Salmonella* contamination of carcasses was greater for slaughter pigs with *Salmonella* infection in lymph nodes compared to the pigs with un-infected lymph nodes. This result strongly evidences the link between on-farm status and carcass contamination risk. Nevertheless, when carcass contamination was compared in slaughterhouses with similar inputs of infected pig lymph nodes, differences were also observed, a fact which was repeated by other contemporary studies involving different slaughterhouses [89,92,95,96,97]. The result demonstrates that the impact of *Salmonella* inputs by infected animals in carcass contamination varies among establishments, indisputably associated with hygiene in the carcass fabrication.

And what about the third and last example? Well, on the one hand, it demonstrates that if we fail in *Salmonella* control in post-harvest, non-infected pigs become positive, but also that correct carcass processing, from a hygienic perspective, can mitigate external carcass contamination, even in highly infected batches.

The preceding section provided a brief overview of various control options that can be implemented on farms. While interventions on farms lack standardization, slaughterhouses have adhered to a standardized procedure for decades to uphold hygiene and ensure meat safety through the hazard analysis critical control points (HACCP) system. This system serves as a framework for monitoring the food system, aimed at reducing the risk of foodborne illness. Different points in the slaughter process or carcass fabrication can be incorporated into the HACCP plan of the abattoir. The control of endemic bacteria, including *Salmonella* can be performed through proper cleaning and disinfection with good manufacturing practices rules. According to Borch and colleagues [98], the following affiliation to CCPs made for specific steps during slaughter and dressing may serve as a guidance: (i) lairage (CP), (ii) killing (CP), (iii) scalding (CP), (iv) dehairing (CP), (v) singeing/flaming (CP), (vi) polishing (CP), (vii) circumoral incision and removal of the intestines (CCP), (viii) excision of the tongue, pharynx, and in particular the tonsils (CCP), (ix) splitting (CP), (x) post mortem inspection procedures (CCP) and (xi) deboning of the head (CCP). Most, if not all, of these CCPs have been highlighted by *Salmonella* post-harvest studies, but the most important CCPs associated with *Salmonella* contamination and spread, both between carcasses and within the slaughter environment, include carcass singeing, carcass scalding (considering water temperature and duration of scalding), carcass inspection (to detect the presence of fecal material), lacerations at evisceration [39,58] or carcass chilling [99,100]. Additionally, strategies such as hot water showers [18] and showers with antibacterial compounds (e.g., organic acids or disinfectants like peroxides) aid in mitigating carcass contamination at the conclusion of the fabrication process [48,101]. Individual plans in the slaughterhouse should identify and correct their CCPs and guarantee that strategies that help in *Salmonella* mitigation (singeing, potential hot washing or chilling) are run accordingly.

Apart from the strategies integrated into the HACCP plan, other measures contributing to *Salmonella* control involve enhancing hygiene in the slaughter process, encompassing both the environment and equipment. Moreover, mitigating the risk of cross-contamination during human carcass dressing activities is imperative, emphasizing strict hygiene practices such as tool sterilization (e.g., knives) and frequent glove changes [39,48]. Furthermore, interventions targeting the processing stage should be complemented by actions to limit the risk of new infections during transport and lairage, as discussed in previous reviews [39]. Enhancing cleaning protocols in both transport and lairage [64] and restricting transport duration or distance and minimizing lairage resting time are well-known practices effective in reducing *Salmonella* transmission risk.

## 7. *Salmonella* Control and Legal Aspects

The main objective of *Salmonella* control in livestock species is mitigating the risk for humans. The current legislation, actions taken or lack thereof have consequences in the aim just exposed. For the first time in this review, let us talk about *Salmonella* control in poultry within the EU. There are currently compulsory programs to control the bacteria in chickens, laying hens and breeders both in poultry and turkey meat production [102]. These control programs include microbiological samplings with *Salmonella* isolation, a fact that together with the zoonoses monitoring in humans [103] offers accurate information on the serotypes and strains circulating both in humans and animals and their associations. In the hypothetical case of human outbreaks, molecular typing methods allow us to trace back the case to the infection source [104,105]. The lack of bacteriological analyses in control programs, or even worse, the lack of control programs and involvement of some of the production stages in the actions taken to control *Salmonella* in swine, hampers clarifying and effectively cutting the outbreaks from swine origin. As an example, in 2015 and 2016, two outbreaks of the emerging monophasic variant of *Salmonella* Typhimurium occurred in the state of Washington in the United States [106,107]. Both outbreaks occurred by the consumption of roasted infected pork and involved over 40 human infections with hospitalizations [107]. According to the scientist involved in the investigations of the first outbreak in 2015 “*Our investigation could not determine the relative importance of specific points in the pork production process that contributed to this outbreak*” and “*we were unable to assess practices or conduct environmental or animal testing at establishment A’s source farms because farms were reluctant to participate, and unclear jurisdictional authority of state agriculture agencies did not require farms to comply with our request*”. Indeed, Salmonellosis is not listed as an animal reportable disease in the US despite its proven importance, which can involve large morbidity and relevant mortality rates in humans [108], a fact that limits potential investigations in live animals. Another recent outbreak in another country with no control program does not offer information about the source of infection either [109]. While in the references provided above [104,105], outbreaks could be traced back, the lack of official involvement of producers in official *Salmonella* surveillance and control in pigs [106,109] was an obstacle to disclosing the origin of both outbreaks. Furthermore, in the outbreak detected in the US, an investigation could prevent a second outbreak with pigs from the same origin occurring in 2016. The examples provided here clearly point out the importance of control programs and the involvement of all production stages in the mitigation of the first foodborne zoonosis worldwide [110]. When authorities excuse any of the actors from the *Salmonella* equation, for instance, primary production on the farm, despite their relevance in the equation, demonstrated by the information gathered here or elsewhere [47,48,51,56], commitment in the mitigation of the human risk fails, as pointed out by Kawami and colleagues [106].

## 8. Conclusions

Over the last 20 years, actions have been put in place to mitigate the main foodborne zoonosis in industrialized countries, which, together with antimicrobial resistance and slurry environmental pollution, are the major One Health problems to be tackled by the swine industry in the forthcoming years. The European Union led the initiative with consolidated compulsory programs in avian species which have ended up in half of the human Salmonellosis reported two decades ago. The success and progress in avian production contrast with the slow implementation of actions in the pig industry. A higher complexity of production systems, infection epidemiology, and lack of clear strategies to mitigate *Salmonella* have discouraged its control, with countries that have not implemented any control and others that have discontinued it. Unfortunately, outside the EU, there is not any leading country in *Salmonella* mitigation in pigs and pork, despite the frequent outbreaks observed and the evidenced risk for humans. Monitoring and surveillance programs offer valuable information to mitigate the pre-harvest and post-harvest *Salmonella* hazards. Countries with programs in place should, in our opinion, put in place actions to reduce prevalence both in animals and pork but avoid penalty systems as much as possible, which limit the profitability of the production and envisage a negative reaction to deal with *Salmonella* but also finding a strategy which does not waste the economic efforts of monitoring and control programs. In addition, the scientific knowledge and experience running control programs acquired through the last decades are useful to re-think and design new, where not in place, efficient and reliable programs to mitigate the pathogen, always considering economic constraints. As stated above, it is highly important to involve and encourage the participation of all the production stages and build awareness of the impact that health actions taken in primary production have on human and environmental health, named One Health.

## Figures and Tables

**Figure 1 animals-14-01035-f001:**
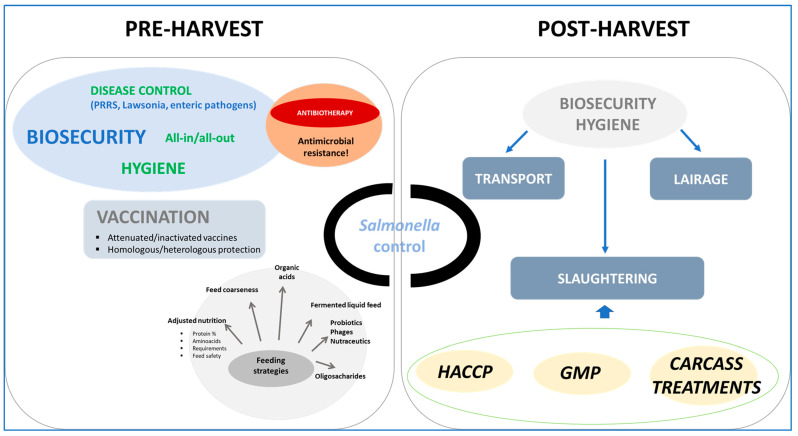
Summary of *Salmonella* control options in pre-harvest and post-harvest swine production.

**Table 1 animals-14-01035-t001:** Summary of surveillance programs in place since the 1990s in *Salmonella* control in pigs.

Program (by Country)	Status	Farm Monitoring	Carcass Monitoring	Penalties	Demonstrated Impact	References
Denmark	Ongoing	Yes	Yes	Yes	Yes	[13,17,18]
Germany	Ongoing, voluntarily	Yes	No	Yes	No	[15,21]
Ireland	Ongoing, under animal health program	Yes	No	No	No	[14]
United Kingdom	Discontinued	Yes	No	No	No	[16,22]
Belgium	Ongoing, voluntarily	Yes	No	No	No	[24]
The Netherlands	Ongoing	Yes	No	No	No	[23]

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
