# Peer review of "Salmonella Control in Swine: A Thoughtful Discussion of the Pre- and Post-Harvest Control Approaches in Industrialized Countries"

_animals, 2024, doi:10.3390/ani14071035_

Round 1

Reviewer 1 Report

Comments and Suggestions for Authors The article provides updated information in reference to the different Salmonella control points. However, it does not delve much into the strategies used in the aforementioned HACCP programs or their effect on the reduction of Salmonella spread at abbatoir. More discussion in this section would be interesting, and would have resulted in a more innovative article. In any case, the review is very well documented and interesting to read.

Author Response

Please,

See attached the response

Thanks

Reviewer 2 Report

Comments and Suggestions for Authors

Animals-2905293 comments for authors

This is an interesting and informative article for the general reader interested in the significant public health issue of salmonella food poisoning and also for researchers and policy makers for whom this is a relevant topic.  The authors have considerable relevant experience and provide useful direction to detailed reviews of particular aspects of the subject, in addition to informative discussion points.

In many ways this paper is a hybrid of a review and an opinion piece and the authors need to provide more clarity on their reasons for writing the paper.   Although some of the reasons are provided in the abstract and summary, a fuller description and justification of the aims is required in the paper itself.  The sentence at lines 55-57 should be expanded.  Was the purpose to evaluate findings and the success or otherwise of control programmes rather than to summarize the literature?  The frequent references to other review articles suggests this.  Did you wish to identify gaps in knowledge or were you highlighting weaknesses in the approach of control programmes as adopted in different jurisdictions?  There should also be a description of how the review was conducted; how the literature search was done and the sources to be included decided upon? 

It is stated in the summary that the authors will suggest future directions that should be adopted in salmonella control but this aspect of the paper is not addressed in a substantive way.

Specific comments:

Line 108.  For clarity this should read:  All other European……………………based on serological Salmonella……

Line 124 and Table 1: It would be useful to clarify that the Competent Authorities and Food Business Operators conduct carcass monitoring in other European countries and although this monitoring is not part of the control programmes, the results may help to inform the success of the programmes.

Line 142:  “Yields” might be better replaced with “Results” or “Outcome”

Line 169 – 2012, not 2015?

Line 184:  This is an important point and the complexity of reasons for lack of farmer engagement with Salmonella control and failure to introduce penalty or incentive schemes by industry or governments in many countries is worth further exploration.  Discussion of behavioural science studies in this area and justification or otherwise for more such studies would be useful, especially in light of the fact that so much is known about Salmonella control in pigs but limited (if any) progress has been made at farm level (even in Denmark). This point arises again at lines 261-265 where the authors list many interventions for control on farm but do not cite any references to support their statements.  Are any intervention studies published that specifically evaluate implementation of several measures simultaneously for Salmonella control?

Lines 291-293.  The authors should explain that the reason for this finding was probable cross-contamination at the abattoir and thus the findings reported are not so surprising.  This is an important clarification to ensure this statement is not misleading for the reader who does not consult the original source (reference 88).

Lines 302-303.  It could also be argued that there would not be such high levels of contamination if pigs from highly infected farms did not introduce contamination in the first place!

Lines 397-400.  Unfortunately, I do not find this concluding sentence convincing as the authors have not provided ideas or offered an opinion as to why or how they think continued monitoring and surveillance will work?  Or how to design efficient and reliable programmes?  Why not just give up (as the UK has done)?    The authors have provided a very useful review and offered some of their experience and insights into this very complex problem but if there is no easy solution then that should be stated rather than suggesting that a solution is within reach through monitoring etc.

Line 314 – this sentence is incomplete.

Comments on the Quality of English Language

Lines 269, 327, 334 – fabrication is not used in this context in English.  Carcass dressing?

As English is not the first language of the authors, some corrections to sentence construction and grammar are required.  In addition, careful proof-reading, including of references and rewording of some colloquial expressions (‘have chosen’ at line 36, ‘probably’ in lines 147 and 178, ‘with 4 legs’ at line 219, ‘lets talk about’ at line 352) are required.

Author Response

(The authors gave the same response as above.)

Reviewer 3 Report

Comments and Suggestions for Authors

Dear authors,

the manuscript entitled  "Salmonella control in swine: a thoughtful discussion of the pre-  and post-harvest control approaches in industrialized countries" is review article which represent critical analysis of Salmonella control programs in developed countries and their efficacy.

This manuscript can serve as a source of valuable data regarding Salmonella control programs in countries with intensive pig production.

Only few comments are given bellow.

Best regards

Reviewer

Suggestions and comments:

Line 28: I suggest to add one more keyword, for example “Salmonella”

Line 32-33: Salmonella enterica subspecies enterica

Line 35: S. Choleraesuis

Line 162 and 164: Please put space between Salmonella Typhiumurium and DT104

Line 239-240: I suggest to write: Again, different vaccines have been tested using different approaches (74-77), but they are not…

Line 286-287: contaminated carcasses

Line 296: colleagues

Line 306: all over the world

Figure 1: Antibiotherapy

Comments on the Quality of English Language

Only minor editing is required as stated above.

Author Response

(The authors gave the same response as above.)
